# Effect of Oral Supplementation of Healthy Pregnant Sows with Sucrosomial Ferric Pyrophosphate on Maternal Iron Status and Hepatic Iron Stores in Newborn Piglets

**DOI:** 10.3390/ani10071113

**Published:** 2020-06-29

**Authors:** Rafał Mazgaj, Mateusz Szudzik, Paweł Lipiński, Aneta Jończy, Ewa Smuda, Marian Kamyczek, Beata Cieślak, Dorine Swinkels, Małgorzata Lenartowicz, Rafał R. Starzyński

**Affiliations:** 1Department of Molecular Biology, Institute of Genetics and Animal Biotechnology, PAS, 05-552 Jastrzębiec, Poland; r.mazgaj@ighz.pl (R.M.); m.szudzik@ighz.pl (M.S.); a.jonczy@ighz.pl (A.J.); e.smuda@ighz.pl (E.S.); 2Pig Hybridization Centre, National Research Institute of Animal Production, Pawłowice 64-122, Poland; marian.kamyczek@zdpawlowice.pl; 3LabSoft Ltd, 02-844 Warsaw, Poland; bc@labsoft.pl; 4Department of Laboratory Medicine (TLM 830), Radboud University Nijmegen Medical Center, 6525 GA Nijmegen, The Netherlands; dorine.swinkels@radboudumc.nl; 5Hepcidin Analysis, Department of Laboratory Medicine, Radboud University Medical Center, 6525 GA Nijmegen, The Netherlands; 6Department of Genetics and Evolutionism, Institute of Zoology and Biomedical Research, Jagiellonian University, 30-387 Kraków, Poland; malgorzata.lenartowicz@uj.edu.pl

**Keywords:** sucrosomial ferric pyrophosphate, iron deficiency anemia, pig, pregnancy, iron supplementation

## Abstract

**Simple Summary:**

In most mammals, including humans, the need for iron increases rapidly in the last period of pregnancy. Therefore, in compliance with World Health Organization (WHO) recommendations, iron supplementation has become a standard procedure even in healthy pregnant women although it carries the risk of iron toxicity and dysregulation of systemic iron homeostasis. Due to physiological and genomic similarities between swine and humans, pigs constitute an useful animal model in nutritional studies during pregnancy. Here, healthy pregnant sows were supplemented with sucrosomial ferric pyrophosphate (SFP), a new non-heme iron formulation, to study its effect on their iron metabolism and that of their progeny. In particular, we aimed at verifying whether supplementation of pregnant sows with SFP will increase the level of low hepatic iron stores in newborn piglets. Results of our study show that SFP does not significantly alter neither systemic iron homeostasis in pregnant sows, nor hepatic iron stores in newborn piglets, which can be used during neonatal period for the maintenance of hematological status. We hypothesize that supplemental iron given orally to pregnant sows is poorly transferred across the placenta.

**Abstract:**

Background: The similarities between swine and humans in physiological and genomic patterns, as well as significant correlation in size and anatomy, make pigs an useful animal model in nutritional studies during pregnancy. In humans and pigs iron needs exponentially increase during the last trimester of pregnancy, mainly due to increased red blood cell mass. Insufficient iron supply during gestation may be responsible for the occurrence of maternal iron deficiency anemia and decreased iron status in neonates. On the other hand, preventive iron supplementation of non-anemic mothers may be of potential risk due to iron toxicity. Several different regimens of iron supplementation have been applied during pregnancy. The majority of oral iron supplementations routinely applied to pregnant sows provide inorganic, non-heme iron compounds, which exhibit low bioavailability and intestinal side effects. The aim of this study was to check, using pig as an animal model, the effect of sucrosomial ferric pyrophosphate (SFP), a new non-heme iron formulation on maternal and neonate iron and hematological status, placental transport and pregnancy outcome; Methods: Fifteen non-anemic pregnant sows were recruited to the experiment at day 80 of pregnancy and randomized into the non-supplemented group (control; n = 5) and two groups receiving oral iron supplementation—sows given sucrosomial ferric pyrophosphate, 60 mg Fe/day (SFP; n = 5) (SiderAL^®^, Pisa, Italy) and sows given ferrous sulfate 60 mg Fe/day (Gambit, Kutno, Poland) (FeSO_4_; n = 5) up to delivery (around day 117). Biological samples were collected from maternal and piglet blood, placenta and piglet tissues. In addition, data on pregnancy outcome were recorded.; Results: Results of our study show that both iron supplements do not alter neither systemic iron homeostasis in pregnant sows nor their hematological status at the end of pregnancy. Moreover, we did not detect any changes of iron content in the milk and colostrum of iron supplemented sows in comparison to controls. Neonatal iron status of piglets from iron supplemented sows was not improved compared with the progeny of control females. No statistically significant differences were found in average piglets weight and number of piglets per litter between animals from experimental groups. The placental expression of iron transporters varied depending on the iron supplement.

## 1. Introduction

In humans, maternal and fetal iron needs exponentially accelerate during the last trimester of pregnancy [1]. Most of gestational iron demand results from the increase in maternal red blood cell (RBC) mass and placental and fetal growth [2]. To meet iron requirements during pregnancy, both absorption of dietary iron and mobilization of this microelement from hepatic stores are enhanced. Iron deficiency during pregnancy is associated with growth retardation, premature birth, low birth weight, muscle dysfunction and low physical capacity [3,4,5,6]. Even in developed countries many women enter pregnancy with insufficient iron stores and dietary iron intake during pregnancy remains consistently below nutritional recommendations [7]. Nowadays, the World Health Organization (WHO) recommends daily iron supplementation for all pregnant women [8]. However, it is questionable whether iron supplements given to healthy, non-anemic women may improve maternal iron status (concentration of iron in colostrum, milk and hepatic iron stores), influence fetus development and neonatal Apgar score [9,10].

The pig is being increasingly used in biomedical research for studies of human diseases that are not accurately represented by rodent models [11,12,13]. For example, the pig model of neonatal iron deficiency anemia (IDA) meticulously reflects the main etiological factor of this defect observed in pre-term human neonates, that is, those having critically low iron content in their livers [14]. Since the molecular potential of iron uptake from the diet in neonates is greatly reduced [15] hepatic iron stores established through maternal-fetal transfer represents the primary source of this microelement to cope with the metabolic demands of developing organisms in the neonatal period. In both, pig term and human preterm neonates insufficient initial iron stores are considered a primary and probably most important etiological factor in the development of neonatal IDA. However, while in preterm human neonates the shortage of stored iron results from shortened period of iron deposition in the fetal liver, in term newborn piglets the main reason is the physiological inability of pregnant sow to meet iron demand for the greater number of fetuses. Several studies have attempted to increase the level of iron hepatic iron stores in fetuses by treating pregnant sows with iron supplements [16,17,18,19,20,21,22]. However, supplementation of sows at various stages of pregnancy, using various iron supplements administered orally or parenterally has no significant impact on the improvement of the iron status of newborn piglets and thus does not prevent suckling animals from becoming anemic (reviewed in Reference [23]).

Recently, liposomal, Sucrosomial^®^ technology became a powerful and promising new formula of sucrosomial ferric pyrophosphate (SFP), non-heme iron characterized by increased bioavailability and reduced toxicity [24,25]. SFP represents an innovative oral iron-containing carrier in which ferric pyrophosphate is protected by a phospholipid bilayer membrane mainly from sunflower lecithin and sucrester matrix [26]. Sucrester is a surfactant derived from the esterification of fatty acids with sucrose (sucrose esters), which has recently been shown to behave as absorption enhancer, because of its ability to reduce intestinal barrier resistance. So far, very promising experiment regarding efficacy of SFP supplementation during human pregnancy has been performed [27].

This study was conducted to determine whether daily oral supplementation of healthy pregnant sows with SFP containing 60 mg Fe/kg of feed during pregnancy is a safe and effective procedure, improving iron status of sows and assuring a rise in the content of iron in hepatic stores in their offspring. Considering that mechanisms of iron transfer across the placenta are far from being fully elucidated in pigs, in our study we also aimed at investigating pathways of iron trafficking across the placenta. Finally, we evaluated pregnancy outcomes in SFP supplemented sows compared with females given ferrous iron sulfate (FeSO_4_) and non-supplemented controls.

## 2. Materials and Methods

### 2.1. Sows and Piglets, Experimental Design and Biological Sample Collection

The experiment was conducted at the Pig Hybridization Centre in Pawłowice belonging to the National Research Institute of Animal Production (Balice, Poland). As shown in Figure 1, total of 15 healthy, non-anemic pregnant 990 line sows (according to veterinary examination and hematological indices shown in Table 1.) and their offspring (sows were mate with the same boar) housed in standard conditions (70% humidity and a temperature of 22 °C in cages with straw bedding) were used. The sows during gestation were in gestation cage (dimensions 2.2 × 0.65 × 1.8 m). Sows were taken to the farrowing cages at 110^th^ day of gestation (dimensions 2.4 × 3.4 m). A gestation crate is used for pregnant sows, which can effectively prevent them from fighting for food, biting and is conducive to sow miscarriage, Sows were fed individually within these cages and at gestation sows were fed 3.0 kg/day of fed given in two portions and water was available ad libitum. Sows used in experiment were in second parity order and had an average weight of 213.90 ± 22.49 kg. Sows were randomly allotted to control and 2 iron supplemented groups. Control females were offered until delivery a standard fodder for pregnant sows routinely used in swine industry (containing 80 mg Fe/1 kg as estimated by flame spectrometry). Fodder was designed to fulfill the National Research Council (NRC) [27] iron requirements for pregnant sows (Appendix A). In the first experimental group, sows were fed with standard fodder and starting from day 80 of pregnancy up to delivery were orally supplemented with additional iron in the form of sucrosomial ferric pyrophosphate (SFP) (SiderAL^®^, PharmaNutra, Pisa, Italy) and given in the amount of 60 mg Fe daily. In the second experimental group, pregnant sows were supplemented with iron in the form of ferrous sulfate (FeSO_4_, Gambit, Kutno, Poland) added to the standard fodder and given to sows according to the same timing and dosage as in the SFP group. Blood from sows was drawn on days 80 and 115 of gestation by venipuncture of the jugular vein (*Vena jugularis externa*). Blood and tissue samples from piglets were collected 24 hours after birth. Piglets were euthanized by intracardiac injection of 0.5 mL/kg body weight of Morbital^®^ (133.3 mg/mL of sodium pentobarbital + 26.7 mg/mL of pentobarbital; Biowet, Puławy, Poland). The blood samples from sows and piglets were collected into tubes coated with heparin as an anticoagulant, centrifuged (1200× *g*, 10 min, 4°C) to separate the plasma. Plasma samples were immediately aliquoted and stored at −80 °C. Tissue samples collected from piglets for molecular analyses were rinsed with PBS and snap frozen in liquid nitrogen then stored at −80 °C. Placenta samples were collected immediately after parturition, snap frozen in liquid nitrogen for molecular analyses, other placenta samples were fixed with paraformaldehyde for immunofluorescence analyses. Colostrum samples were collected immediately after delivery and milk samples 48h after delivery, both were stored at −20 °C until further analysis.

### 2.2. Measurement of Red Blood Cell Indices and Plasma Iron Level

Hematological indices were determined using an automated ADVIA 2010 analyzer (Siemens, Erlangen, Germany). The plasma iron concentration was determined by colorimetric measurement of an iron-chromazurol complex according to the manufacturer’s protocol (Biomaxima S.A., Lublin, Poland) as previously described [28].

### 2.3. Measurement of Non-Heme Iron Content in Tissues

The non-heme iron content of liver, spleen (100 mg wet tissue; wt) and milk and colostrum (1 mL) were determined by acid digestion of the samples at 100 °C for 10 min, followed by colorimetric measurement of the absorbance of the iron-ferrozine complex at 560 nm as previously described [29].

### 2.4. Plasma Hepcidin-25 Measurement

Piglet plasma hepcidin-25 measurements were performed by a combination of weak cation exchange chromatography and time-of-flight mass spectrometry (WCX-TOF MS), as described previously for pig plasma [30] and urine [31]. Peptide spectra were generated on a Microflex LT matrix-enhanced laser desorption/ionization TOF MS platform (Bruker Daltonics, Billerica, MA, USA).

### 2.5. Real-Time Quantitative RT-PCR

Total cellular RNA was extracted from liver and placental tissue (20 mg) using Trizol reagent (Invitrogen) according to the manufacturer’s protocol. Two micrograms of total DNAse-treated RNA were reverse transcribed using a Transcriptor First Strand cDNA Synthesis Kit (Roche Diagnostics, Mannheim, Germany). Real-time quantitative polymerase chain reaction (PCR) analysis was performed in a Light Cycler U96 (Roche Diagnostics, Mannheim, Germany) using gene-specific primer pairs (Appendix A. The amplified products were detected using SYBR Green I (Roche Diagnostics, Mannheim, Germany) as described previously [32]. To confirm amplification specificity, the PCR products were subjected to melting curve analysis and agarose gel electrophoresis. Light Cycler U96 Software (Roche Diagnostics, Mannheim, Germany) was used for data analysis. Transcript levels were normalized relative to the control reference gene selected using NormFinder software (Aarhus, Denmark) [33] (https://moma.dk/normfinder-software).

### 2.6. Immunofluorescence (IF) Analysis and Confocal Microscopy of Placental Sections

After delivery, pig placentas were immediately dissected and fixed in 4% paraformaldehyde (Sigma-Aldrich, Poznan, Poland) in phosphate-buffered saline (PBS) at 4 °C for 24 hours. Following two 30 min washes in PBS, the tissues were successively soaked in 12.5 and 25% sucrose (Bioshop, Burlington, Ontario, ON, Canada) for 24 h and 7 days, respectively, at 4 °C. Placenta was embedded in Cryomatrix medium (Thermo Fisher Scientific, Warsaw, Poland), frozen in liquid nitrogen and sectioned in 10-μm slices using a cryomicrotome (Shandon, London, UK).

The sections were washed in PBS for 10 min and permeabilized by bathing in PBS/0.1% Triton X-100 (Sigma-Aldrich, Poznan, Poland) for 20 min. Non-specific antibody binding was blocked by incubating the tissue sections in PBS/5% Bovine Serum Albumin (BSA) (Bioshop, Burlington, Ontario, ON, Canada) for 1 h. For protein detection, sections were incubated overnight at room temperature with primary antibodies diluted in PBS/5% BSA. As a negative control, some sections were incubated without primary antibody. The primary and fluorochrome-conjugated secondary antibodies used in IF analysis are described in Appendix A. Next, the sections were washed for 5 × 6 min with PBS/0.1% Triton X-100 and incubated for 1 h with secondary antibody diluted in PBS/5% BSA at RT. Finally, sections were washed for 5 × 6 min in PBS and additionally stained with Hoechst (Thermo Fisher Scientific, Warsaw, Poland) for 2 min then washed 2 times with PBS and mounted in glycerol based medium. The antibodies used can be found in Appendix A.

### 2.7. Placental Samples and Fixation for Transmission Electron Microscopy (TEM)

Five placentas per experimental group were obtained from sows following normal, uncomplicated births and pregnancies. Small pieces of basal plate (10 × 10 × 2 mm) were excised within minutes of delivery (expelled placentas) and fixed for 1 h in 4% paraformaldehyde, 0.4% glutaraldehyde in sodium cacodylate trihydrate (Acros organics, Geel, Belgium) pH 7.2, followed by 1% OsO_4_ (Thermo Fisher, Kandel, Germany) for 16 h in 4 °C. Samples were then dehydrated through an ethanol series (30; 50; 70%). Samples were dried using Leica EM CPD300 Critical Point Dryer (Leica, Wetzlar, Germany) and sputtered with gold using Low Vacuum Coater Leica EM ACE200 (Leica, Wetzlar, Germany). Transmission electron microscopy was conducted using COXEM EM-30AXplus SEM (Yuseong-gu, Daejeon, Korea) microscope. Transmission Electron Microscopy (TEM) images can be found in Appendix A.

### 2.8. Ethic Statement

The experimental procedures used in this study were in compliance with the European Union (EU) guidelines for the care and handling of research animals (EU Directive2010/63/EU for animal experiments). Second Local Ethical Committee on Animal Testing at the Warsaw University of Life Sciences in Warsaw (Poland) granted a formal waiver of the ethical approval because the only procedure involved in the study was piglets euthanasia and blood collection from pregnant sows. All these procedures are the routine veterinary procedures. Moreover, administration of a dietary supplement such as sucrosomial ferric pyrophosphate (SFP) is not categorized as a research procedure.

### 2.9. Statistical Analysis

Statistical analysis of iron supplementation in one-day-old piglets was performed using one-way analysis of variance (ANOVA) with type of supplementation as the factor followed by Tukey’s multiple comparisons test. The results from sows with iron supplementation were calculated by two-way ANOVA, using type of supplementation and time as the factors followed by Sidak’s multiple comparisons test. Statistical analysis and figures were prepared using GraphPad Prism version 8.00 for Windows (GraphPad Software, La Jolla, CA, USA). *p*-value, *p* ≤ 0.05 and *p* ≤ 0.01 were considered significant and are denoted with asterisks * and ** respectively.

## 3. Results

### 3.1. Preganant Sows Supplemented with SFP and FeSO_4_ Show No Changes in Hematological and Iron Status Compared with Non-Suplemented Controls

At the starting point of the experiment, that is, on day 80 of pregnancy, values of red blood cell (RBC) indices measured in all sows involved in the study were similar (Table 1). Likewise, sows from different experimental groups showed no differences in plasma iron parameters at this stage of pregnancy (Table 1). As pregnancy progressed (day 114) RBC indices remained unchanged with the exception of some fluctuations in RBC count and hemoglobin level in FeSO_4_ supplemented sows. In contrast, in advanced pregnancy, a concerted, statistically significant decrease in plasma iron, ferritin and hepcidin levels was observed in sows from all experimental groups. Changes in these parameters indicate decrease in iron stores in sows from all experimental groups (regardless of iron supplementation) at the end of pregnancy. At the same time, on day 114 of pregnancy there were no differences in plasma iron status between control and iron supplemented sows.

### 3.2. Supplementation of Pregnant Sows with Iron Did Not Influence Iron Content in the Colostrum and Milk

To verify whether supplementation of pregnant sows with iron increases the concentration of this microelement in the milk and colostrum, we evaluated this parameter in sows from all experimental groups. Iron content in these biological fluids was not significantly affected by oral administration of neither SFP nor FeSO_4_ to pregnant sows compared with non-treated females (Figure 2). Although a slight up regulation of iron content in supplemented groups was observed but it is was not statistically significant.

### 3.3. Red Blood Cell Indices and Plasma Iron Status in Piglets from Control and Iron-Supplemented Sows

Results shown in Table 2. clearly demonstrate that supplementation of pregnant sows either with SFP or FeSO_4_ did not improve RBC indices and plasma iron status of their piglets. Importantly, on day 1 after birth hemoglobin concentration in all newborn piglets was above the threshold value of anemia in this range of age, 8 g/dL) [33].

### 3.4. Supplementation of Pregnant Sows with SFP or FeSO_4_ Does Not Alter neither Iron Content in the Placenta nor Hepatic and Splenic Iron Status of their Progeny

Importantly in context to pig farming and fattening, the number of piglets in the litter remains unchanged regardless of iron supplementation procedure in comparison to untreated sows. Similarly, piglets from sows receiving different iron supplementations or without iron treatment showed an equal body weight, measured one day after delivery (Appendix A). Iron transfer across the placenta is the only pathway providing this microelement from mother to developing fetuses. Although non-heme iron content in the placenta of sows supplemented with SFP or FeSO_4_ showed a downward trend compared with control females, no statistically significant differences were found (Figure 3). This indicates that despite administration to sows of various iron supplements, iron flux through the placental barrier remained similar in sows from all experimental groups. This is reflected in almost equal iron accumulation in piglets’ liver and spleen, key organs for handling systemic iron (Figure 3).

### 3.5. Increased Hepcidin and Bone Morphogenetic Protein 6 (BMP6) Expression in the Placenta and in the Liver of Piglets from Sows Supplemented with SFP

Hepcidin, the master regulator of systemic iron metabolism, is mainly synthesized and released by hepatocytes in response to increased body iron concentration [34,35,36,37,38,39,40,41]. Hepcidin expression has been also reported in many other mammalian cells including placenta [42,43,44,45,46,47,48,49]. Although the function of tissue-specific hepcidin remains unknown it has been suggested that local autocrine regulation based on locally produced hepcidin may operate in several tissues [50]. To evaluate the effect of oral administration of pregnant sows with SFP on hepcidin expression, we measured hepcidin mRNA abundance in the liver of piglets and in the placenta. In tissues collected form the group of SFP-treated sows, hepcidin expression was increased in the liver (statistically significant difference) and in the placenta (a strong upward trend) compared with tissues obtained from control animals (Figure 4). In contrast to the effect of SFP on hepcidin hepatic mRNA expression in piglets, no induction of hepcidin mRNA expression was observed in livers from piglets derived from mothers supplemented with FeSO_4_ (Figure 4). To check whether hepatic expressional pattern of hepcidin is reflected in the level of hepcidin peptide circulating in the blood, we measured the concentration of active hepcidin-25 in the blood plasma of piglets. We found a similar trend in plasma hepcidin concentration, that is, strong increase in piglets from SFP-treated sows compared with controls and basic level in piglets from pregnant sows supplemented with FeSO_4_ (Figure 4).

Bone morphogenetic protein 6 (BMP6) is a central regulatory factor that increases hepatic hepcidin expression in response to iron. Sinusoidal hepatic endothelial cells are the predominant source of BMP6 in the liver, which acts in a paracrine manner by binding to complex BMP6 receptor on hepatocytes to induce hepcidin transcription [51]. Therefore, we attempted to answer whether BMP6 is involved in hepcidin regulation in the placenta and in the liver of 1-day old piglets after iron supplementation of sows. We found that in both tissues expression pattern of BMP6 perfectly overlaps that of hepcidin in animals from all analyzed groups. This strongly suggests the involvement of BMP6 in the regulation of hepcidin expression under our experimental conditions.

### 3.6. The Effect of Oral Iron Supplementation on Placental Morphology and Expression of Iron Transporters

The placenta serves as the interface between mother and fetus and it mediates nutrient transport to the fetus including the transport of iron. Uni-directional transfer of iron transported by the placenta from the maternal to the fetal circulation is effected by transferrin receptor 1 (TfR1), divalent metal transporter 1 (DMT1) and ferroportin (Fpn), located on the apical and basolateral membrane of syncytiotrophoblats, respectively [52]. To determine the influence of supplementation of pregnant sows with SFP on iron transporters expression and localization in the placenta, we analyzed mRNA abundance of TfR1, DMT1 and Fpn as well as localization of these three proteins responsible for iron flow across the placenta. Administration of SFP to pregnant sows had no significant effect on the level of transcripts encoding TfR1, DMT1 and Fpn compared with controls. In contrast, we observed a downward tendency in the expression of analyzed genes in the placenta from sows supplemented with FeSO_4_. In the case of DMT1, this downregulation was statistically significant compared to controls (Figure 5A). We also showed that in placentas from three experimental groups principal transporters mediating cellular iron uptake and efflux are abundantly and equally expressed in syncytiotrophoblasts (STB) on the maternal and fetal sides, respectively (Figure 5B–D). This indicates that the efficiency of iron transport across the placentas from sows from all groups is similar.

To check the potential effect of SFP on the integrity and morphology of placenta we performed an exhaustive analysis of placenta using scanning electron microscopy at different scanning resolutions. We did not observe any visible morphological damages/changes at the surface of syncytiotrophoblasts collected from sows either supplemented with SFP or FeSO_4_ compared with controls. (Appendix A). This result is consistent with previous studies indicating low toxicity SFP [24,53,54].

## 4. Discussion 

The pig is an interesting experimental animal model for studying iron supplementation during pregnancy. While pregnant sows of most contemporary pig breeds are usually iron replete and do not manifest symptoms of iron deficiency, their progeny regularly develops IDA approximately 3–4 days after birth [23,55,56,57]. The fundamental cause of neonatal anemia in pigs is a drastic imbalance between poor iron supply and high iron demand. Poor availability of iron in newborn piglets occurs due to extremely low level of hepatic iron stores (the lowest in mammalian neonates) [14,29,55,58,59,60,61,62] and low iron content in the colostrum/milk [63], accompanied by inefficient absorption of dietary iron [55]. On the other hand, high iron needs are determined by unusually rapid rate growth of piglets (they increase 10-fold their body mass within 6 weeks of birth) [15,55,60,61,64]. The concept of replacing routine, largely non-physiological postnatal parenteral supplementation of piglets with iron dextran [65] by administration of iron to pregnant sows to prevent suckling animals from becoming anemic has emerged in several studies [66]. The rationale behind such a procedure is to increase iron status of pregnant females, intensify iron transfer across the placenta from the mother to the fetuses, consequently increasing hepatic iron content, which can serve as a source of this microelement in piglets during early postnatal development. Among potential benefits of this treatment include reduction of the labor input and improving the welfare of supplemented young animals. However, this treatment is highly challenging considering its possible adverse effects on sow’s iron metabolism, the risk of iron toxicity and insufficiency of the molecular machinery involved in transplacental iron transport. Indeed, supplementation of sows at various stages of pregnancy, using various iron supplements (iron salts/chelates, iron dextran) administered orally or parenterally has no significant impact on the improvement of iron status of newly born piglets and has not been proven in prophylaxis of neonatal IDA in piglets [16,19,20,21,67,68,69,70,71,72]. Despite these negative results, in this study we attempted to test the efficacy of prenatal oral supplementation with SFP through administration of this compound to pregnant sows. SFP is an innovative preparation of ferric pyrophosphate, covered by phospholipids plus sucrester matrix, with high bioavailability, capacity to overcome gastrointestinal barriers and tolerability [23,24,25,26,53,54]. Importantly, the efficacy and no side effects of SFP administration have been reported in pregnant women with iron deficiency [26]. Here, we show that despite supplementation of sows from day 80 of pregnancy with SFP, their iron status few days before delivery was decreased similarly to sows supplemented with FeSO_4_ or fed with control diet. However, this moderate decrease did not compromise RBC status of pregnant females. An uniform drop in ferritin plasma levels (in sows from all experimental groups), a marker of hepatic iron stores [73], strongly indicates that to maintain erythropoietic activity of pregnant females, iron is preferentially released from the liver. Similarly, decrease in the concentration of plasma hepcidin observed in sows from all groups on day 114 of pregnancy results from, at least, partial depletion of iron stores. It is known that under conditions of enhanced erythropoiesis observed in late pregnancy [74,75,76], hepcidin is down-regulated independently from iron deficiency by erythroid factors produced by erythroblasts that act on hepatocytes to suppress hepcidin synthesis [77]. Indeed, it has been reported that not only anemic mothers [77] but also mothers with iron-replete stores [78] show low hepcidin expression at delivery [79]. The possible explanation for hepcidin suppression in pregnant females (even in those supplemented with iron) is the need to increase endogenous maternal iron supply for extra gestational requirements by enhanced iron mobilization from stores, increased iron absorption and accelerated recycling of iron derived from senescent erythrocytes.

Most importantly for this study, supplementation of sows with SFP failed to reinforce iron stores in newborn piglets. All iron indexes such as hepatic and splenic iron content, blood plasma iron parameters attest ineffectiveness of SFP supplementation of sows to improve iron status in piglets. These results collectively with aligned non-anemic RBC pattern of piglets from all experimental groups, confirm previous reports demonstrating that 1-day old piglets born from sows either supplemented with iron preparations or not, do not yet show symptoms of IDA [47]. We hypothesize that the failure in the reinforcement of piglets iron status in response to SFP administration to pregnant sows is associated with the limited molecular potential of the placenta to increase iron transport from the mother to fetuses. Immunofluorescent analysis of 3 main iron transporters such as TfR1, DMT1 and Fpn clearly shows their analogous distribution and similar intensity of fluorescent signal in placentas from all experimental groups. Accordingly, loading pregnant mice with iron showed no influence on the protein level of TfR1 and Fpn in the E18.5 placenta compared with the placenta from females fed standard iron diet [78]. Surprisingly, we noticed that administration of SFP to pregnant sows consequently induced hepcidin expression in the placenta (increase in the mRNA level) and in piglets (increase in the hepatic mRNA level and in the concentration of circulating peptide in the blood) and overlapped the rise in the expression of placental and hepatic BMP6, factor that increases hepatic hepcidin in response to iron [34]. This set of data strongly suggest the involvement of discreetly increased signaling (regulatory) intratissular iron pool, which was not detectable using our analytical methods. Up-regulation of hepcidin may be a part of control mechanisms protecting fetuses and newborns from excessive iron transport through ferroportin from the placenta to the fetal circulation and therefore from exacerbated iron toxicity associated with iron overload [80]. Alternatively, it is likely a consequence of the ancestral regulation of iron transport across the placenta functioning in wild boar. In contrast to domestic pig progeny, hepatic iron stores in wild boar piglets are adequate to meet iron needs for erythropoiesis probably because during pregnancy iron transferred from the wild boar mother has to be distributed among only 4–6 fetuses [81] instead of 10–14 in domestic pig sows of high performance contemporary breeds.

## 5. Conclusions

In conclusion, it seems that SFP is an efficient iron supplement only in iron deficient subjects [25,26,53,54] including iron deficient pregnant mothers as it has been demonstrated in humans [26]. When pregnant females are iron replete or show slightly decreased iron status as is the case of pregnant sows, supplementation with SFP is inefficient. The mechanism of prevention of excessive iron accumulation in the body upon treatment with SFP in purpose to increase iron stores is an interesting challenge for future research. Taken together, the results of this study demonstrate the effectiveness of daily oral dose of 60 mg SFP on occurrence and prevention of anemia during swine pregnancy. In perspective the research challenge is to use oral iron supplementation to treat IDA during pregnancy in a pig model of maternal anemia. The importance of iron for feto-maternal health and fetal development during pregnancy cannot be overestimated. Therefore the question of how iron is transported from mother to fetus is still open. There are still candidates for placental iron transport, such as ZIP8, ZIP14 or FLVCRa and b or heme iron transporters, whose altered expression may be associated with fetal or maternal iron status [82,83]. One should not forget about the possibilities of non-transferrin bound iron (NTBI) transport through the placenta. Such iron appears in the blood mostly in hemochromatic patients when saturation of transferrin exceeds 70% or 80%. There are also evidence that in pregnant woman taking iron supplements NTBI may rise [84]. Thus future research should take into account both the above-mentioned transporters as well as should explain the mechanisms of iron transport across the placenta and their effect on the regulation of iron metabolism during healthy and complicated pregnancy.

## Figures and Tables

**Figure 1 animals-10-01113-f001:**
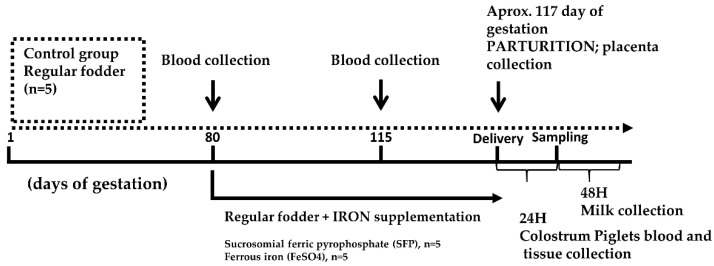
Experimental design scheme. Pregnant sows were allotted to 3 groups: (i) control group receiving a standard fodder; (ii) group supplemented with ferrous iron sulfate (FeSO_4_); (iii) group supplemented with sucrosomial ferric pyrophosphate (SFP). Both supplementations were applied daily between day 80 of pregnancy and delivery (approximately day 117 of pregnancy). Blood samples were collected from sows on days 80 and 115 of pregnancy. Samples of placenta were collected immediately after delivery. Tissue and blood samples from piglets were collected 24 h after birth.

**Figure 2 animals-10-01113-f002:**
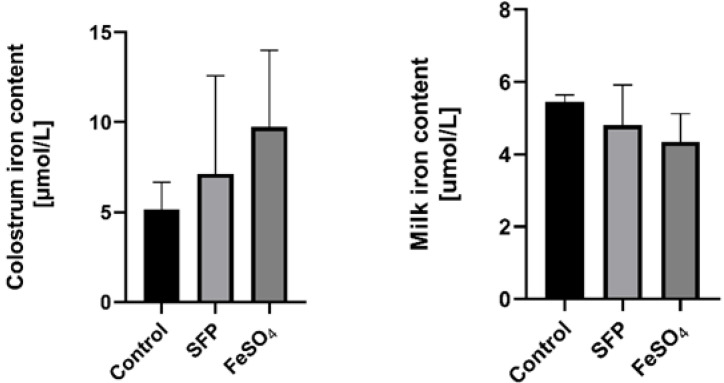
Effect of ferrous iron sulfate (FeSO_4_)and sucrosomial ferric pyrophosphate (SFP) oral supplementation of pregnant sows on milk and colostrum iron concentrations. The non-heme iron content was measured in colostrum and milk as described in the Materials and Methods section. Values are expressed as the means ± S.D. for 1 mL samples obtained from sows immediately (colostrum) and 24 h after delivery (milk) (n = 15).

**Figure 3 animals-10-01113-f003:**
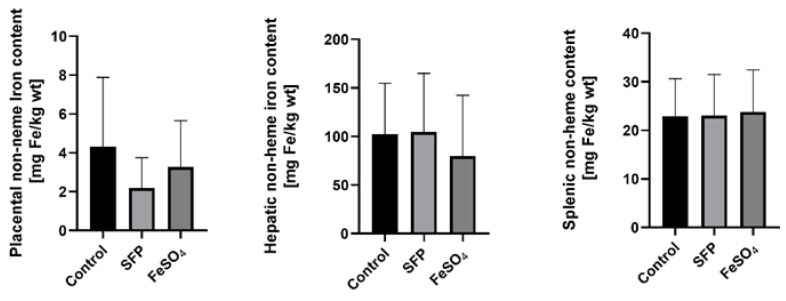
Non-heme iron content in placenta from sows after ferrous iron sulfate (FeSO_4_) and sucrosomial ferric pyrophosphate (SFP) supplementation and hepatic and splenic non-heme iron content from 1-day old piglets. Non-heme iron content in analyzed tissue was measured as described in Materials and Methods. Values are expressed as the mean and SD (standard deviation) for samples obtained from 9 piglets in each experimental group. Non-heme iron content was expressed as mg Fe/kg of wet tissue.

**Figure 4 animals-10-01113-f004:**
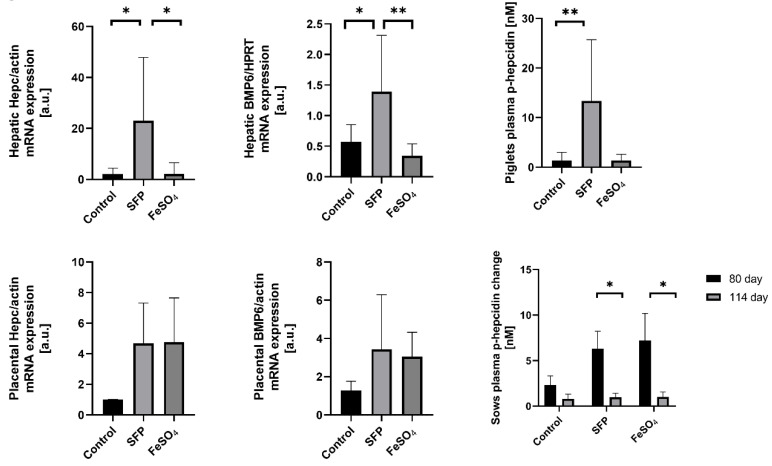
Changes in placental and fetal hepatic and plasma hepcidin and Bone Morphogenetic Protein 6 (BMP6) levels under different oral iron supplementations. Fetal hepatic and placental hepcidin and BMP6 mRNAs levels were measured using Real Time polymerase chain reaction (PCR) and normalized to actin/Hypoxanthine Phosphoribosyltransferase (HPRT) mRNA. Hepcidin-25 (p-hepcidin) measurements in sows and piglets plasma were performed by peptide enrichment through weak cation exchange chromatography coupled to time-of-flight mass spectrometry (WCX-TOF MS) (Bruker Daltonics, Billerica, Massachusetts, United States) [29,30]. * and ** asterisks denote statistically significant differences between parameters in control and ferrous iron sulfate (FeSO_4_) or sucrosomial ferric pyrophosphate (SFP) supplemented group at *p* < 0.05 and *p* < 0.01 respectively.

**Figure 5 animals-10-01113-f005:**
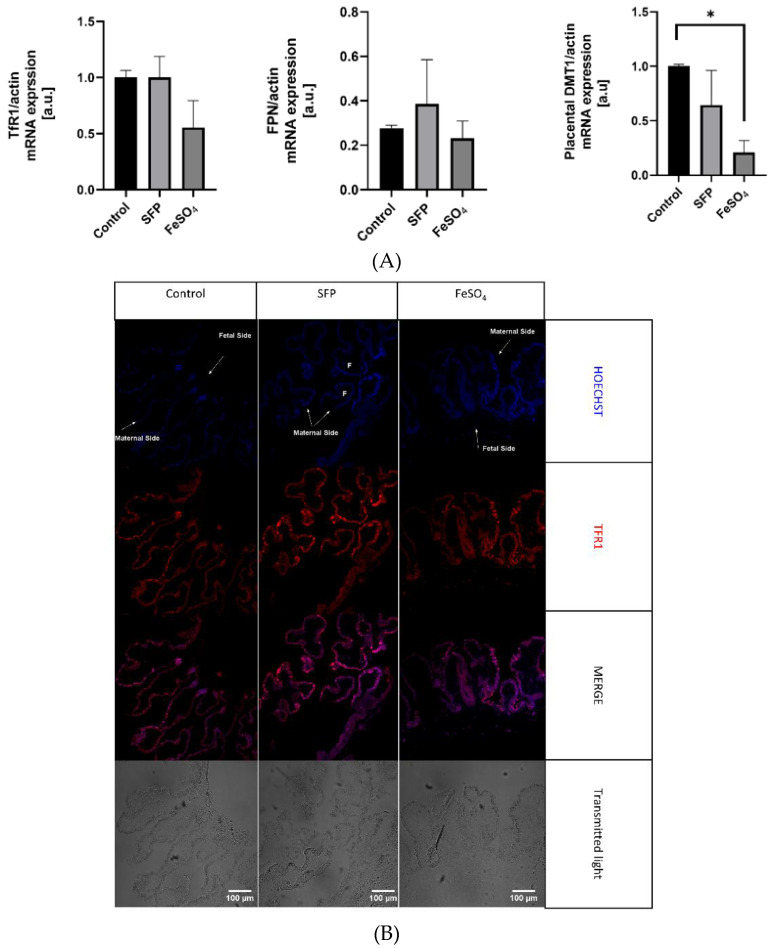
(**A**). Regulation of placental iron transporters after oral administration of ferrous iron sulfate (FeSO_4_) or sucrosomial ferric pyrophosphate (SFP) to pregnant sows. Placental iron transporters mRNA expression measured using Real time PCR normalized to actin mRNA. * asterisk denote statistically significant differences between parameters in control and SFP or FeSO_4_ supplemented group at *p* < 0.05. (**B**). Localization of placental iron transporters after oral administration of ferrous iron sulfate (FeSO_4_) or sucrosomial ferric pyrophosphate (SFP) to pregnant sows. Representative immunofluorescence staining for placental iron transporters: TfR1. Scale bars correspond to = 100µm. Cell nuclei were counterstained with Hoechst (blue). Arrows indicate maternal or fetal site of syncytiotrophoblast (STB). Series sections of placental tissue from 15 sows was analyzed and representative immunofluorescence photos were prepared. (**C**). Localization of placental iron transporters after oral administration of ferrous iron sulfate (FeSO_4_) or sucrosomial ferric pyrophosphate (SFP) to sows. Representative immunofluorescence staining for placental iron transporters: DMT1. Scale bars correspond to = 100µm. Cell nuclei were counterstained with Hoechst (blue). (**D**). Localization of placental iron transporters after oral administration of ferrous iron sulfate (FeSO4) or sucrosomial ferric pyrophosphate (SFP) to pregnant sows. Representative immunofluorescence staining for placental iron transporters: Fpn. Scale bars correspond to = 100µm. Cell nuclei were counterstained with Hoechst (blue).

**Table 1 animals-10-01113-t001:** Red blood cell (RBC) indices and plasma iron status in sows at days 80 and 114 of pregnancy.

Experimental Groups	Gestational Days		Supplementation		Time		Supplementation * Time		*p*-Value 80 vs. 114 Day
	Day 80	Day 114							
			F	*p*-Value	F	*p*-Value	F	*p*-Value	
	**Hb** (g/dL)								
Control	12.2 ± 0.8	11.7 ± 1.9	6.539	0.0311^*^	0.1384	0.7227	0.5869	0.5851	0.9519
SFP	9.8 ± 2.9	11.0 ± 0.6							0.9993
FeSO_4_	11.1 ± 0.88	10.2 ± 2.24							0.7189
	**RBC** (mln/mm^3^)								
Control	6.2 ± 0.6	5.8 ± 1.1	6.480	0.0317^*^	1.504	0.2660	0.5245	0.6167	>0.9999
SFP	5.3 ± 1.7	5.6 ± 0.3							0.8496
FeSO_4_	5.7 ± 0.6	6.9 ± 4.6							0.5106
	**MCV** (µm^3^)								
Control	61.2 ± 3.1	63.2 ± 2.7	0.2633	0.7769	38.23	0.0008 ^**^	1.409	0.3150	0.192
SFP	58.4 ± 4.9	62.8 ± 3.5							0.0132^*^
FeSO_4_	62.4 ± 4.9	64.5 ± 4.5							0.0198^*^
	**Plasma iron level** (µmol/L)								
Control	15.6 ± 3	12.3 ± 4.3	0.02066	0.9796	4.767	0.0717	0.1594	0.8562	0.8294
SFP	14.4 ± 4.7	12.8 ± 2.8							0.4135
FeSO_4_	15.3 ± 1.8	12.8 ± 1.3							0.5222
	**Plasma ferritin concentration** (ng/mL)								
Control	514.3 ± 117.4	199.8 ± 57	0.8294	0.4808	29.20	0.0017^**^	0.7431	0.5148	0.0597
SFP	557.9 ± 67.5	231.1 ± 56.4							0.0218^*^
FeSO_4_	402.7 ± 180.4	176.4 ± 85.3							0.1831
	**Plasma hepcidin level** (nM)								
Control	2.33 ± 0.8	0.8 ± 0.42	3.979	0.0794	41.62	0.0007 ^**^	4.506	0.0638	0.5571
SFP	6.3 ± 1.58	0.96 ± 0.36							0.0115 ^*^
FeSO_4_	7.2 ± 2.43	1 ± 0.45							0.0055 ^**^

Data are presented as mean values ± SD. Statistical analysis of two factors have been performed by two-way ANOVA for repeated measurement. Two factors analyzed in two-way ANOVA were “Time,” “Supplementation” and their interaction. SFP = sucrosomial ferric pyrophosphate, FeSO_4_ = ferrous iron sulfate RBC = red blood cell count, Hb = hemoglobin level, MCV = mean corpuscular volume, F = variance of the group means/mean of the within group variances. All parameters were determined for 5 sows from each experimental group. * and ** asterisks denote statistically significant differences at *p* < 0.05 and *p* < 0.01.

**Table 2 animals-10-01113-t002:** Red blood cell indices and plasma iron status in 1-day old piglets from control and SFP and FeSO_4_-supplemented sows.

Blood Parameters	Control	SFP	FeSO_4_	*p*-Value	
				Control *vs.* ….	
Hb (g/dL)	9.4 ± 1.4	9.8 ± 1	9 ± 1.3	SFP 0.6906	FeSO_4_ 0.7161
RBC mln/mm^3^	4.9 ± 0.7	5.1 ± 0.9	4.69 ± 0.6	SFP 0.8252	FeSO_4_ 0.7332
MCV (µm^3^)	60 ± 2.7	63.1 ± 3.7	61.6 ± 4	SFP 0.1301	FeSO_4_ 0.5538
Serum Iron (µmol/L)	14.8 ± 9.9	10.5 ± 8.4	18 ± 8.1	SFP 0.2284	FeSO_4_ 0.9994
Serum Ferritin (ng/mL)	823.5 ± 161.4	906.2 ± 75.1	772.7 ± 137.6	SFP 0.3656	FeSO_4_ 0.6631

Effect of ferrous iron sulfate (FeSO_4_) and sucrosomial ferric pyrophosphate (SFP) oral supplementation on blood parameters in one day old piglets. RBC—red blood cell count, Hb—hemoglobin level, MCV—mean corpuscular volume. All parameters were determined for 5 sows from each experimental group. n = 9 piglets per experimental group.

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
