# Peer review of "Effect of Oral Supplementation of Healthy Pregnant Sows with Sucrosomial Ferric Pyrophosphate on Maternal Iron Status and Hepatic Iron Stores in Newborn Piglets"

_animals, 2020, doi:10.3390/ani10071113_

Round 1

Reviewer 1 Report

The manuscript, #animals-832741, by Rafał Mazgaj et al. describes the experiment to determine the effects of oral supplementation of healthy pregnant sows with sucrosomial ferric pyrophosphate on maternal iron status and hepatic iron stores in newborn piglets. The topic shows meaningful data in Fe nutrition and Fe additives both on sow and most similarly on woman during late pregnancy. The manuscript is relatively well written and easy to read, which consider to be further published after minor revision.

General comments

-The effects of two kinds of Fe supplementation were discussed in this article. However, the basis of Fe in control diet is not listed and this may influence the Fe status of sows and piglets.

-Each treatment has five replicates in this experiment. Are they enough to reflect the effect of Fe supplementation as sows often show high individual differences?

-Authors need to add information in detail on sow cage size, feeding individually or grouped, and sow feeding regime as well.

-In the Materials and Methods section, please add the information of sow body weight and sow parity.

Specific comments

Line 120, describe how to perform the oral supplementation of SFP, and how to confirm the SEP being totally ingested by sows.

Line 117-118, the authors indicated that “Fodder was designed to fulfill the NRC iron requirements for pregnant and nursing sows (Supplementary Table 1)”, but there is not a nursing sow diet yet. Why?

Line 135, add space between “at” and “-20°C”

Line 214, add one column to show the p-value on day 80.

Line 410-415, this paragraph is the conclusion of the current article. It seems to be reasonable that this paragraph was moved the Conclusions section.

Line 478-480, the year of “2015” should be bold.

Line 566, the year of “1949” should be bold.

In Supplementary Table 1,

- The creep feeding diet used to be the diet for nursing piglets. Please confirm it.

-Since the value of ME and crude protein is much higher than the NRC requirements for pregnant sows, please double check or explain.

-The concentration of Fe is 120 mg, which is 1.5 times more than the recommendations of NRC 2012. Why?

-Please add data on other micro mineral composition, like Cu, Zn, and Mn.

-Please describe how to get the nutritive value of diet.

-Please describe the feed either as dry basis or as-feed basis

Reviewer 2 Report

In the past, numerous attempts have been made to iron supplement sows but without success. Therefore, the main concept of this study ( to improve iron status in sows and offspring) has a low level of novelty. However, there are other important aspects, which have not been studied before in such experiments. The study on pathways of iron transport across placenta is one of them. Therefore, I recommend this well-written manuscript for publication with some adjustments.

The title says ‘hepatic iron stores’ in newborn piglets, but in fact it should be ‘tissue iron stores’. Splenic iron stores are also measured. In addition, I miss the information about iron transport in the title, which constitutes a large part of this study.

Litter size is an important factor, which should be taken into account while dealing with iron status in sows. Although the average litter size is shown in supplementary Figure, it should be discussed.

Another important aspect, which is missing in this manuscript, is parity of the sows. As hematological status of sows vary with parity, it should be included and discussed.

The initial Hb concentration (at day 80) of SFP sows is much lower compared to control and FeSO4. But the SD is very large in these SFP sows. May be the effect is driven by one or two of these sows? It is not clear if all these sows were non-anemic. Additionally, the criteria of defining the sows as anemic or non-anemic should be mentioned.

Reviewer 3 Report

The manuscript Effect of oral supplementation of healthy pregnant sows with sucrosomial ferric pyrophosphate on maternal iron status and hepatic iron stores in newborn piglets by Mazgaj et al is certainly a nice piece of resesearch. It presents information on the effect o a new and highly available Fe supplement, which is provided to gestating sows. The outcome is observed in the sow, the placenta and in the offspring. Generally speaking, the manuscript confirms previous scientific information and evidence that dietary treatment to sows is quite ineffective in altering piglet iron status.  The manuscript is carried out in a sound manner, with updated methodology and nice discussion. Even thouh results were expected and the main outcome is lack of any effect, I do recommend acceptance for publication.

Comments/Suggestions

Ln 196-200 and Table 1. Please, consider using a repeated measurement statistical test to evaluate the effect of time. This is the recommended statistical procedure when sampling the same animal over the time. It provides the effect of time, the effect of treatment and (more important) the intreraction time * treatment). In this particular Table it can be seen that in some cases (e.g. Hb concentration in plasma) respond differently over the time according to treatment (e.g., while concentration decreases in the control and FeSO4 groups, it actually increases in the 5FP group). This might be of significance and may allow getting some interesting conclusion. In any case, comparison of data from the same animal sampled in two different times cannot be analyzed by ANOVA, as there is a bias in the degrees of freedoum used.
